# Systematic discovery of gene-environment interactions underlying the human plasma proteome in UK Biobank

Robert F. Hillary [1,2,3], Danni A. Gadd[1,2,3], Zhana Kuncheva[1,3,4], Tasos Mangelis[1,3,4], Tinchi Lin[3], Kyle Ferber[3], Helen McLaughlin[3], Heiko Runz[3], Biogen Biobank Team*, Riccardo E. Marioni [1,2,3,6] ✉, Christopher N. Foley [1,3,4,6] ✉ & Benjamin B. Sun [3,5,6] ✉

Understanding how gene-environment interactions (GEIs) influence the circulating proteome could aid in biomarker discovery and validation. The presence of GEIs can be inferred from single nucleotide polymorphisms that associate with phenotypic variability - termed variance quantitative trait loci (vQTLs). Here, vQTL association studies are performed on plasma levels of 1463 proteins in 52,363 UK Biobank participants. A set of 677 independent vQTLs are identified across 568 proteins. They include 67 variants that lack conventional additive main effects on protein levels. Over 1100 GEIs are identified between 101 proteins and 153 environmental exposures. GEI analyses uncover possible mechanisms that explain why 13/67 vQTL-only sites lack corresponding main effects. Additional analyses also highlight how age, sex, epistatic interactions and statistical artefacts may underscore associations between genetic variation and variance heterogeneity. This study establishes the most comprehensive database yet of vQTLs and GEIs for the human proteome.

High-throughput proteomic analyses enable scalable biomarker discovery for complex disease states[1]. A growing number of studies have catalogued genetic influences on the human plasma proteome[2–6]. Sequence variants associated with protein abundances are termed protein quantitative trait loci (or pQTLs) and their colocalisation with disease-associated variants has guided the identification of pathogenic molecular pathways, aiding drug and biomarker validation[7–9]. However, the influences of environmental factors and, in particular, gene-environment interactions (or GEIs) on the human plasma proteome have remained understudied. Determining whether environmental exposures modify genetic associations with protein abundances should provide additional, nuanced insights into protein biology and biomarker discovery.

GEIs most commonly arise when genotype groups at a locus show differential associations between an environmental exposure and a phenotype of interest (e.g. protein levels)[10,11]. There has been relatively limited success in identifying GEIs due to their small effect sizes and challenges in accurately recording multiple environmental exposures over the life course[12,13]. Using all genome-wide genetic variants and hundreds of potential environmental modifiers to test for GEIs also imposes a significant multiple testing burden.

A GEI can manifest in the form of differences in the variance of a given trait across genotypes at a locus (Fig. 1). Therefore, one strategy to infer the presence of a GEI is to perform genome-wide scans for these loci, which are defined as variance quantitative trait loci or vQTLs[14,15]. This is in contrast with pQTLs, which associate with

[1]Optima Partners, Edinburgh EH2 4HQ, UK. [2]Centre for Genomic and Experimental Medicine, Institute of Genetics and Cancer, University of Edinburgh, Edinburgh EH4 2XU, UK. [3]Translational Sciences, Research and Development, Biogen Inc., Cambridge, MA, USA. [4]Bayes Centre, The University of Edinburgh, Edinburgh EH8 9BT, UK. [5]Cardiovascular Epidemiology Unit, Department of Public Health and Primary Care, University of Cambridge, Cambridge CB1 8RN, UK. [6]These authors contributed equally: Riccardo E. Marioni, Christopher N. Foley, Benjamin B. Sun.*A list of authors and their affiliations appears at the end of the paper. ✉e-mail: riccardo.marioni@ed.ac.uk; chris.foley@optimapartners.co.uk; bbsun92@outlook.com

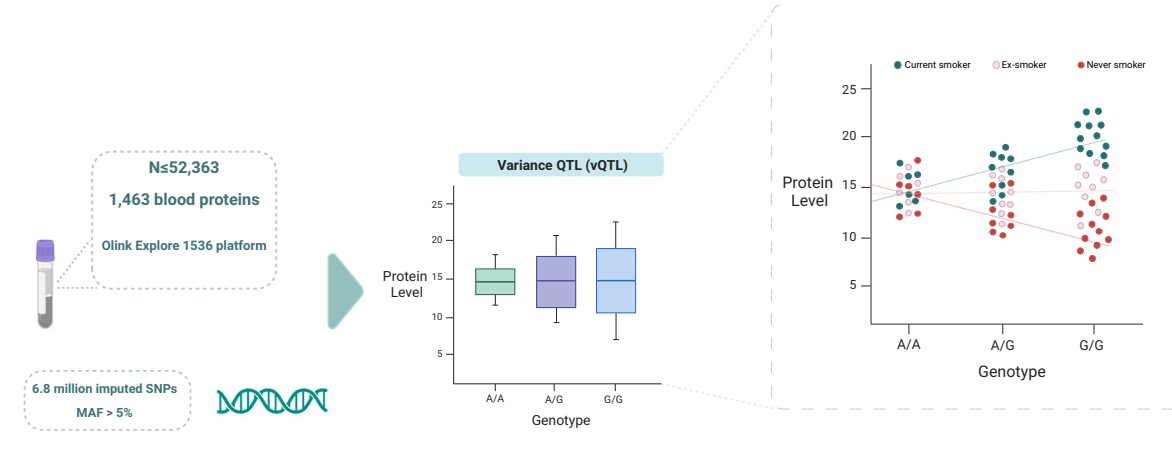

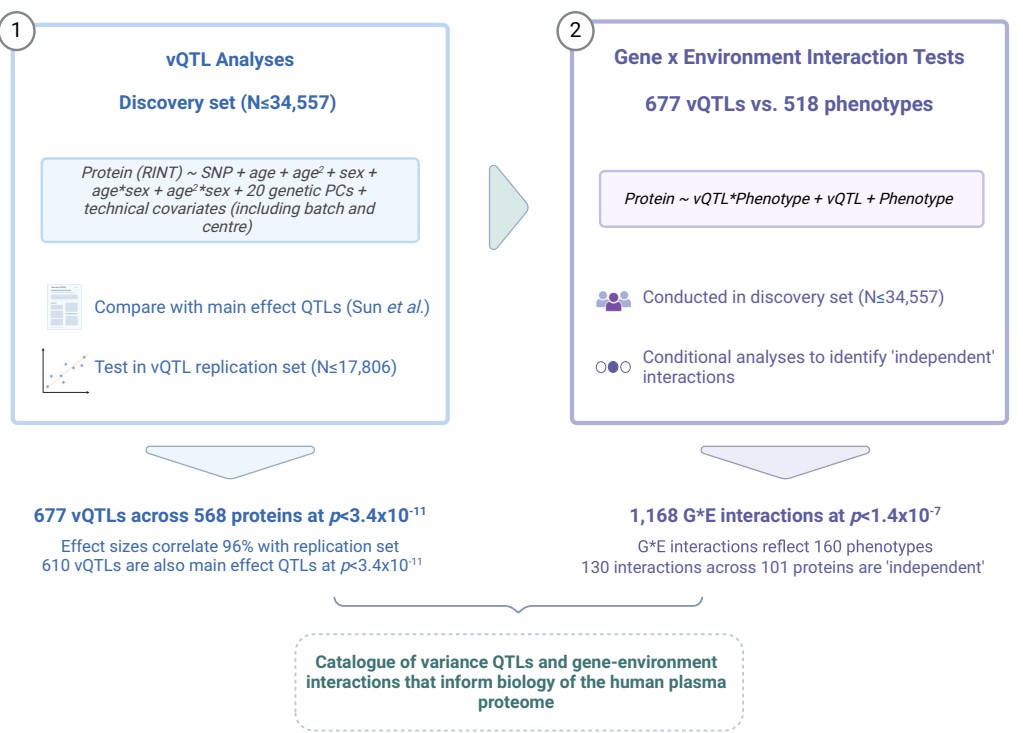

**Fig. 1 | Overview of study design for variance QTL analyses in The UK Biobank Pharma Proteomics Project sample.** Top panel: Plasma levels of 1463 proteins (measured by 1472 Olink analytes) and genotype data were available for up-to-52,363 participants in UK Biobank after quality control. Variance QTL (vQTL) analyses were performed using 6.8 million imputed SNPs to detect loci that associated with differential variances in protein levels across genotypes. Gene-environment interactions (GEIs) can manifest as differences in the variance of a trait (e.g. protein levels) across genotypes at a given polymorphism. In this example, which uses fictitious data, the G-allele positively correlates with protein levels in one sub-group of the sample (current smokers, shown in teal). A negative correlation is observed in never smokers (shown in red). There is no correlation in ex-smokers (shown in peach). Therefore, genotype at this locus interacts with the exposure (e.g. smoking status), which creates a mean-based interaction effect. The effect underlies the dispersion of the data in G-allele carriers and in turn gives the appearance of a vQTL. Bottom panel: The independent discovery and replication sets consisted of 34,557 and 17,806 participants, respectively. Effect sizes and $p$ values for variance QTLs were compared against those from a recent main effect QTL analysis on the same proteins and sample by Sun et al. The use of vQTLs for GEI tests can greatly reduce computational burden. Therefore, we tested whether protein levels were associated with an interaction between their vQTLs and a broad range of health-related phenotypes in UK Biobank. A large number of phenotypes were correlated with one another (e.g. adiposity-related traits). Stepwise conditional analyses were employed in order to identify 'independent' interactions. GEI gene-environment interactions, MAF minor allele frequency, vQTL variance quantitative trait locus. Figure 1 created with BioRender.com released under a Creative Commons Attribution-NonCommercial-NoDerivs 4.0 International license.

differences in mean protein levels across genotype groups. Of note, variants that associate with mean differences in traits (i.e. pQTLs) have been referred to as additive main effect or simply, main effect loci in the vQTL literature[16,17]. Studies have identified vQTLs for lifestyle and cardiopulmonary traits such as blood pressure and body mass index[15,16,18,19]. These studies have also shown that the power to detect

GEIs is enhanced when restricting the genetic search space to vQTLs instead of all genome-wide variants or to QTLs with additive main effects on the outcome (analogous to pQTLs)[15–17]. Westerman et al.[17] applied the two-stage approach of vQTL discovery and GEI testing to serum cardiometabolic biomarkers, which included 10 proteins. However, the vQTL architecture for most human proteins remains

undescribed thereby hindering systematic screens for GEIs that may impact their circulating levels. Identifying vQTLs and associated GEIs in this context could further guide predictions on the safety and efficacy of protein biomarkers and drug targets.

In stage one, we conduct genome-wide vQTL association studies on plasma levels of 1463 Olink proteins in up-to-52,363 UK Biobank participants. In stage two, GEI associations are comprehensively screened for using vQTL loci identified in stage one and over 500 environmental exposures (see Fig. 1 for a summary of the study design). vQTL variants are cross-referenced with a recent pQTL (or main effect QTL) study using the same sample, highlighting aspects of protein biology that may not have otherwise been captured by conventional GWAS models[6]. GEI association tests are also repeated using pQTLs to assess whether there is an enrichment in GEI discovery when restricting the genetic search space to variance effect versus main effect QTLs. We pinpoint environmental factors that explain why some sites affect the variance of protein levels only and fail to show a genetic main effect. We also explore additional explanations for variance heterogeneity, such as epistatic interactions, phantom vQTLs and statistical artefacts from phenotype transformations. This study establishes a comprehensive catalogue of variance effects for the human proteome, which others may utilise to investigate GEIs of interest.

## Results

### Discovery of variance QTLs underlying the plasma proteome
In the first stage of the study, Levene's test with median was used to perform genome-wide vQTL analyses on blood levels of 1463 unique proteins ($N \leq 34{,}557$). The 1463 proteins were measured by 1472 analytes using Olink technology. A Bonferroni-corrected significance threshold was set at $p < 3.4 \times 10^{-11}$, which reflected the adjustment of $p < 5 \times 10^{-8}$ (a commonly used threshold in GWAS)[20] for 1463 proteins. There were 269,225 significant vQTL associations across 575 analytes at $p < 3.4 \times 10^{-11}$. The associations implicated 568 unique proteins. There was limited evidence for genomic inflation (range of $\lambda = [0.9, 1.1]$, Supplementary Data 1). Six hundred and seventy-seven independent vQTLs were identified through linkage disequilibrium (LD) clumping (see "Methods", Supplementary Data 2). Supplementary Data 2 and 3 show genomic annotations of the variants using the Open Targets platform[21,22].

Four hundred and seventy-three (69.9%) of the 677 independent vQTLs were *cis* effects (within 1 Mb from the gene encoding the protein) and the remaining 204 represented *trans* effects (Fig. 2a). The majority of proteins had one independent vQTL (488, 85.9%) and the maximum number of vQTLs per protein was 5 (for FOLR3 and PNLIPRP2, Fig. 2b). There was an inverse relationship between the logarithms of effect sizes and minor allele frequency (MAF) for both *cis* and *trans* loci ($r = -0.45$ and $-0.46$, respectively, MAF $\geq 5\%$, Fig. 2c). One vQTL was in strong LD ($r^2 = 0.87$) with an expression vQTL at the same locus (rs858502 and rs66809776 for *PILRA*, respectively)[23]. However, no protein vQTL overlapped with known DNA methylation vQTLs[24].

Six variants were not available for replication analyses as they had MAF < 5% in the replication set ($N = 17{,}806$). This left 671 variants for replication testing. Effect sizes for the variants were highly correlated between the discovery and replication sets ($r = 0.96$, 95% CI = [0.95, 0.97], Fig. 2d, Supplementary Data 4). Three hundred and ninety-six variants (59.0% of 671) survived Bonferroni correction in the replication set ($p < 3.4 \times 10^{-11}$), and had effect sizes that were directionally concordant with those from the discovery set. Of note, the discovery set had twice the sample size of the replication set. The discovery sample also contained white Europeans only whereas the replication set comprised 80% white Europeans and several other ethnic backgrounds (see Methods). It is likely that differences in ancestries underpinned variants whose effect sizes showed opposing directions across sets. A notable example was the *cis* vQTL rs35489971-A for

CD300LF, which had an effect size of −0.35 in the discovery set and 0.13 in the replication set (highlighted in Fig. 2d). There was evidence for ancestry-specific effects for such outliers. The A allele increased the variance of CD300LF levels in those of Black/Black British ancestry but not in those of South Asian or European ancestries. The A allele is also the major allele in those of Black/Black British ancestry (frequency = 70%) but is the minor allele (<25%) in other ancestries. The ancestry-specific effects likely underscored the positive association between the A allele and CD300LF variance in the replication sample but not the discovery sample, which comprised individuals of European ancestry only. However, sensitivity analyses showed that vQTL effect sizes were largely comparable ($r = 0.82$) between individuals of non-European and European descent when matched for sample size ($N \leq 2518$, Supplementary Data 5). The sample structure was retained, which enabled direct comparisons to a recent main effect QTL study by Sun et al.[6], who implemented the same sampling strategy.

### Variance QTLs largely overlap with main effect QTLs for the blood proteome
The majority of vQTLs (610, 90.1%) had main effect $p$ values < $3.4 \times 10^{-11}$ (Supplementary Data 6). Only fifteen vQTLs lacked a main effect finding at $p < 0.05$. Their $p$-values ranged from 0.08 to 0.98 and three loci were located within the complex *MHC* region. Figure 3a, b demonstrate examples of vQTL loci with and without marginal main effects, respectively. A reverse look-up strategy showed that only 3.4% of main effect QTLs (301 of 8856 variants at MAF $\geq 5\%$) had vQTL $p$ values < $3.4 \times 10^{-11}$. Over 23% of main effect QTLs (2053 of 8856) had vQTL $p$ values below a more relaxed threshold of $p < 0.05$ (Supplementary Data 6). Therefore, most vQTLs were main effect QTLs but not vice versa.

Supplementary Fig. 1 shows the relationship between the significance of *cis* vQTLs or main effect QTLs and their distances from TSS. Figure 3c, d show the predicted functional classes of vQTLs and main effect QTLs. Most predicted classes exhibited little variation between these QTL types given their substantial overlap. However, a higher proportion of main effect QTLs were annotated to exons than vQTLs (17.3% vs. 10.3%). In balance, a lower proportion of main effect QTLs were annotated to intergenic sites when compared to vQTLs (18.9% vs. 28.2%)[6].

Olink assays also rely on limits of detection (LOD) for each protein and sample plate. The percentage of individuals falling below their LOD for each protein is shown in Supplementary Data 1. Individuals were not removed based on LOD in the consortium. It is challenging to select a threshold to filter proteins based on LOD (e.g. removing proteins that had >50% of individuals falling below the LOD). Such thresholds may be arbitrary. We characterised how many vQTL sites were associated with proteins that had an appreciable number of individuals falling below the LOD. Sixty vQTLs were associated with proteins that had >10% of individuals whose relative measurements were below the LOD for that protein. In addition, 39, 13 and 8 vQTLs associated with proteins that had >25%, >50% of >75% of individuals whose measurements did not surpass the LOD. If a variant influenced whether individuals surpassed the LOD, it could introduce disparities in trait variance across carriers and non-carriers. We conducted a sensitivity analysis to assess if vQTL variants were associated with whether individuals were above or below the LOD for their respective proteins (binarised outcome). Forty-one vQTLs (6.1%) were associated with assay detectability $p < 7.4 \times 10^{-5}$ ($p < 0.05$ adjusted for 677 tests, Supplementary Data 7). However, we did not observe a significant difference in LOD distributions between the 575 proteins with vQTL associations when compared to the 895 without ($\chi = 338$, $p = 0.90$). vQTL variants were not associated with missingness due to other technical factors at $p < 7.4 \times 10^{-5}$ (Supplementary Data 8). Therefore, assay detectability, but not other technical considerations, may have exerted an effect on the discovery of a small subset of vQTL associations.

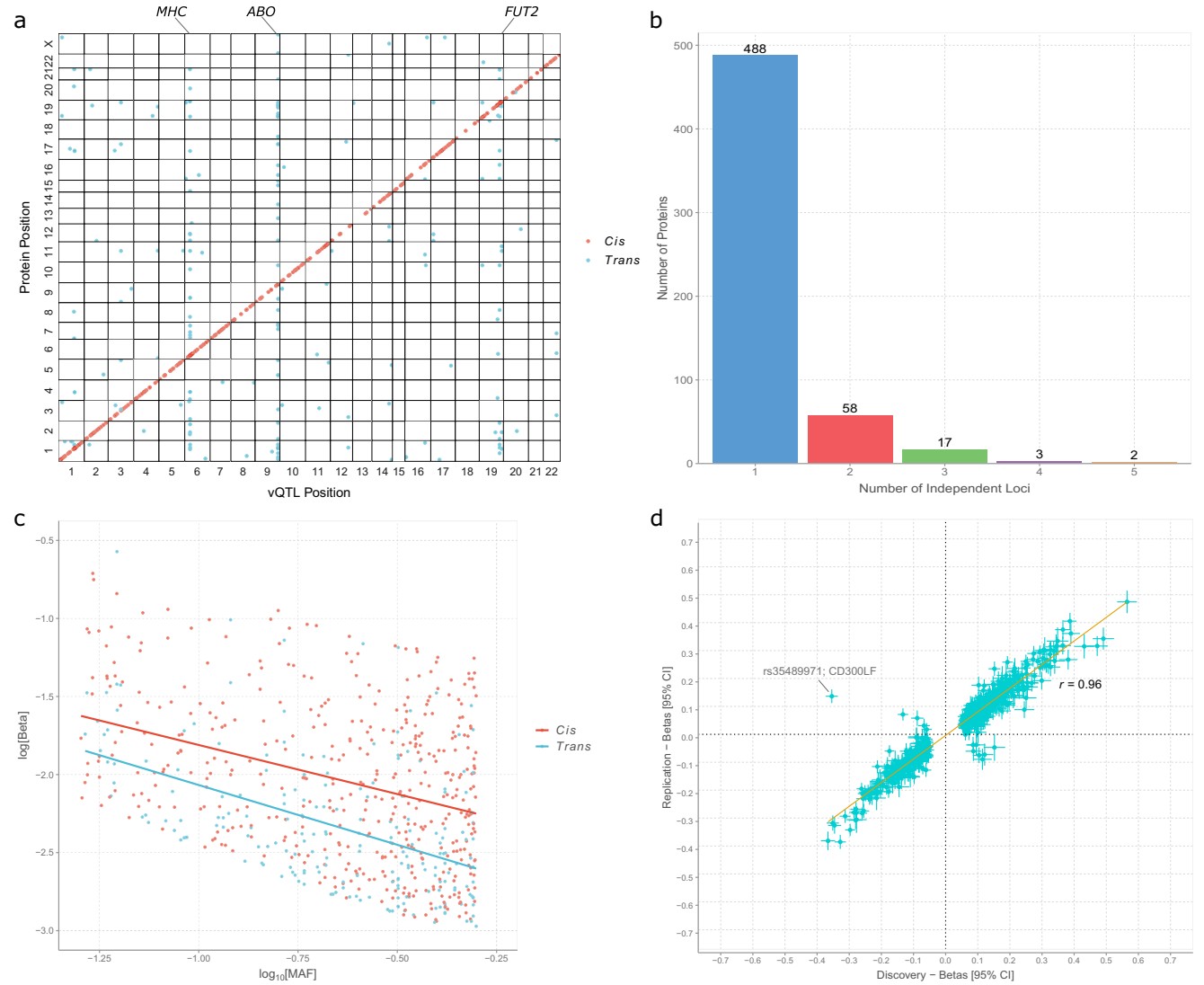

**Fig. 2 | Genome-wide association studies to identify variance QTLs for 1472 blood protein measures in UK Biobank. a** The 1472 analytes or measures represented 1463 unique proteins. Genome-wide variance QTL tests were performed using Levene's test (two-sided). A Bonferroni-corrected significance threshold of $p < 3.4 \times 10^{-11}$ was set. The x-axis represents the chromosomal location of independent *cis* and *trans* vQTLs. The y-axis represents the position of the gene encoding the associated protein. *Cis* (red circles); *trans* (blue circles). **b** The number of independent vQTLs per protein. **c** Association between the common logarithm of minor allele frequencies and the natural logarithm of absolute effect sizes for *cis* and *trans* vQTLs. *Cis* (red circles and line); *trans* (blue circles and line). **d** The discovery and replication sets consisted of 34,557 and 17,806 participants, respectively. Pearson's correlations between effect sizes in the discovery and replication sets are shown for vQTLs that were significant in the discovery set. An outlier is highlighted and resulted from differences in allele frequencies across distinct ethnic groups in the study sample. CI confidence interval, MAF minor allele frequency, vQTL variance quantitative trait locus.

## Variance QTLs unveil gene-environment interactions dispersed across the plasma proteome

Variance QTLs may be explained by a number of potential influences, including gene-gene interactions (epistasis), statistical artefacts and gene-environment interactions. A particular strength of vQTL discovery has rested in nominating a priority set of genetic variants for gene-environment association tests[15,17]. In the second stage of the study, we therefore tested whether protein levels were associated with an interaction between their vQTL(s) and 518 disparate environmental factors (see "Methods" for phenotype selection and quality control criteria). Summary data for these phenotypes are shown in Supplementary Data 9. Briefly, phenotypes were selected through a combination of systematic selection criteria (using the GWAS Catalog[25]) and further manual curation to ensure that a broad range of relevant phenotypes were considered. Variance QTLs were first queried against the GWAS Catalog. Phenotypes that associated with these variants at $p < 5 \times 10^{-8}$ in the

GWAS Catalog were extracted and included in GEI tests if they were also present in the UK Biobank database. Additional continuous phenotypes that were previously examined by Westerman et al.[17] were included, as well as possibly relevant lifestyle or metabolic traits. Possible sources of variance heterogeneity other than GEIs, such as epistasis, phenotype preparation and phantom vQTLs are explored in later sections.

We observed 1168 GEIs at a Bonferroni-adjusted threshold of $p < 1.4 \times 10^{-7}$ when vQTLs were used as index variants ($p < 0.05$ adjusted for 677 variants and 518 exposures). Significant GEIs comprised 15.4% of the vQTLs tested (104 variants) and reflected 101 unique proteins (Supplementary Data 10). The main effect QTL strategy returned 3061 GEIs at the same threshold. However, these associations encompassed only 2.8% of main effect QTLs (244 of 8856 variants, Supplementary Data 11). The proportion of vQTLs that participated in a GEI effect was 5.5-fold higher than the proportion of pQTLs, and this estimate is in line with previous findings[15,17].

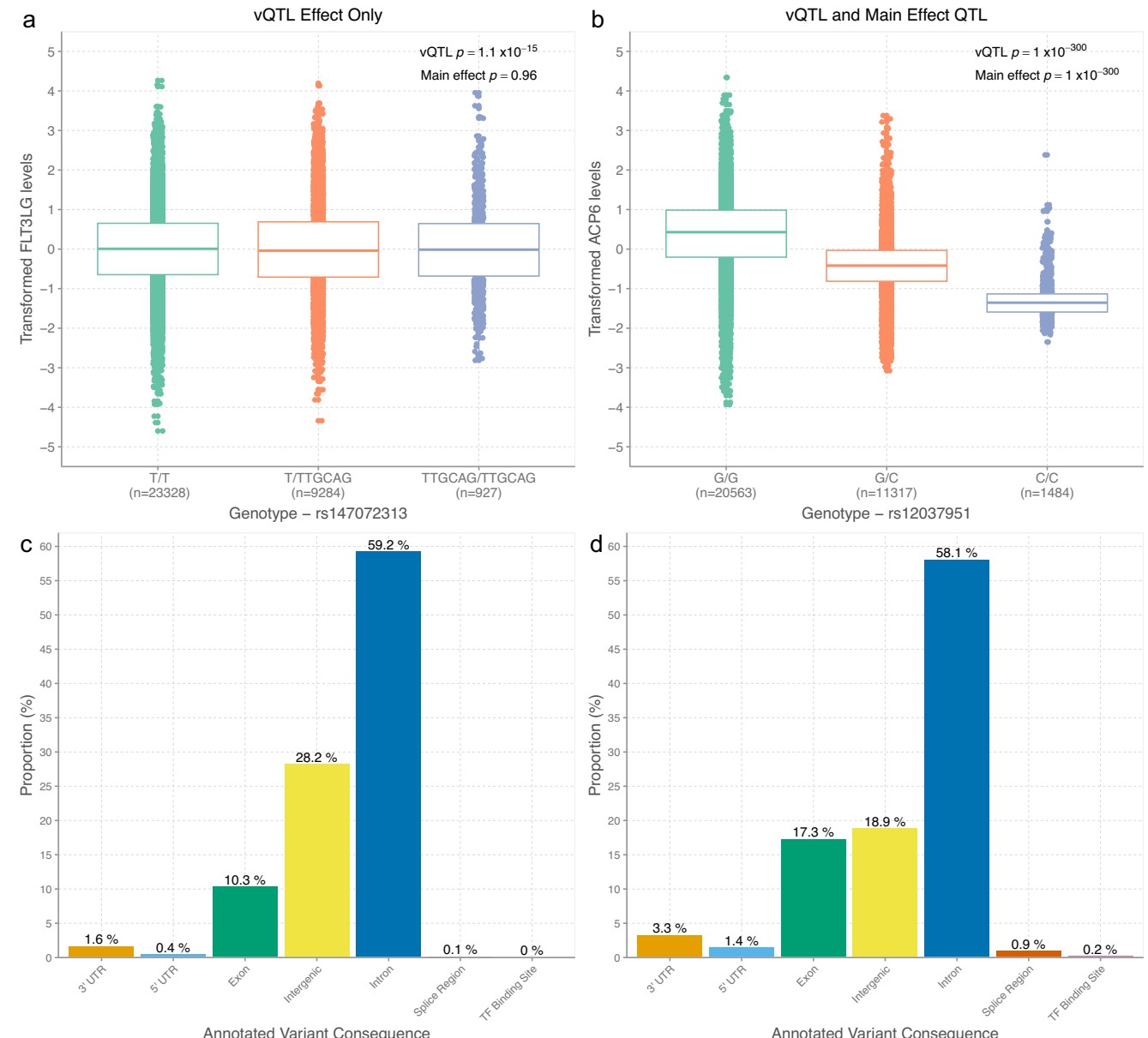

**Fig. 3 | Genetic architectures of main effect and variance QTLs for plasma protein levels. a** Example of variance QTL (rs147072313) without corresponding main effect on protein levels (FLT3LG). Levene's test was used to perform variance QTL tests (two-sided). Centre line of boxplot: median, bounds of box: first and third quartiles and tips of whiskers: minimum and maximum. **b** Example of variance QTL (rs12037951) with significant main effect on protein levels (ACP6). Levene's test was used to perform variance QTL tests (two-sided). Centre line of boxplot: median, bounds of box: first and third quartiles and tips of whiskers: minimum and maximum. **c** Distributions of predicted functional annotation classes for all variance QTLs. Bar height represents the mean proportion of variants within each class. **d** Distributions of predicted functional annotation classes for main effect QTLs. TF transcription factor, UTR untranslated region, vQTL variance quantitative trait locus.

Many of the exposures tested were from highly correlated categories (e.g. physical and lipid traits). Therefore, to identify 'independent' GEI associations, we undertook stepwise conditional analyses for each protein. Here, GEI associations for each protein were iteratively conditioned on the most significant association for that protein. GEIs that failed to survive Bonferroni correction once conditioned on the most significant association were removed. The process was repeated using the next most significant association for that protein until no further phenotypes could be considered (see Methods). These analyses do not point towards biological pathways or potentially underlying biological effects. They were carried out to help account for the correlation structure between GEIs at a given locus. Conditional analyses suggested that 130 of the 1168 GEIs with vQTLs were

'independent' when further accounting for correlations between phenotypes (Supplementary Data 12). Twenty proteins exhibited two or more conditionally significant associations with a maximum of 4 conditional GEIs for FCGR2A, FUCA1 and LDLR (Fig. 4a). Conditional GEIs included 53 unique phenotypes. In total, 115/130 GEIs were nominally significant ($p < 0.05$) and directionally concordant with estimates from the replication set (Supplementary Data 13).

### GEIs implicate known biological pathways and are influenced by age and sex

GEIs revealed a number of interactions consistent with known biology. Thirty-eight conditional GEIs involved phenotypes that were selected via look-up analyses in the GWAS Catalog[25] (Supplementary Data 16).

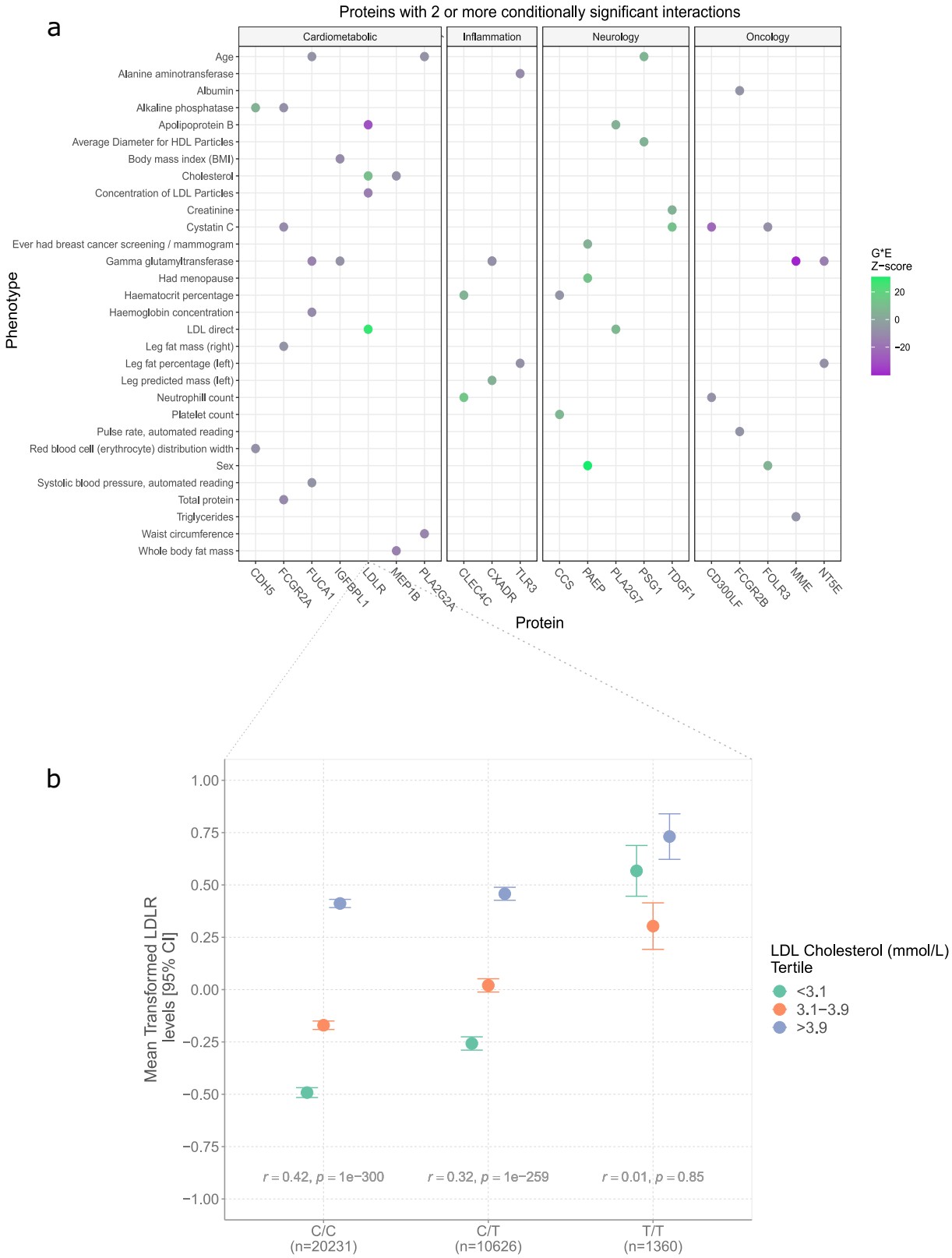

**Fig. 4 | Gene-environment interactions underlying the plasma proteome are pervasive. a** Interaction Z-scores are highlighted only for proteins with two or more conditionally significant GEI effects. Positive Z-scores are shown in green and negative Z-scores are shown in purple. Supplementary Data 14 shows direct associations between protein levels and their respective phenotypes displayed in (**b**) (i.e. not stratified by genotype). **b** Illustrative example: mean transformed LDL receptor or LDLR levels (closed circles) with 95% confidence intervals (vertical bars) when stratified by vQTL genotype (rs75627662) and tertiles of measured blood LDL cholesterol. Data are presented as mean values ±95% confidence intervals. GEI gene-environment interaction, LDL low-density lipoprotein, vQTL variance quantitative trait locus.

These associations entailed variants with known main effects on the exposure (i.e. phenotype) and also on the mean and/or variance of the protein's measured abundance in blood. There were also known correlations between the protein level and phenotype, including an association between LDL receptor (low-density lipoprotein, LDLR) and LDL cholesterol[26] (Fig. 4b). Our GEI analyses supplements existing knowledge by showing genotype-dependent effects for these associations. For instance, the *trans* variant rs75627662 located near *APOE* had an additive effect on transformed LDLR levels and served as a vQTL. There was a strong positive correlation between LDLR and cholesterol concentrations in major (C)-allele carriers ($r_{het}$ = 0.32, 95% CI = [0.31, 0.34], $p = 1.2 \times 10^{-259}$; $r_{major}$ = 0.42, 95% CI = [0.41, 0.43], $p = 1.0 \times 10^{-300}$; Pearson's correlation). The association was ablated in those homozygous for the minor T-allele ($r_{minor}$ = 0.01, 95% CI = [−0.05, 0.06], $p = 0.85$). This underscored a clear GEI that was uncovered by the two-stage strategy. Other GEIs included associations between interleukin-6 and C-reactive protein levels (Supplementary Data 16). Interleukin-6 stimulates the synthesis of C-reactive protein[27]. We also detected a GEI whereby the effect of a *cis* variant on oxytocin levels differed according to sex. Oxytocin shows sex-dependent effects on a range of physical and behavioural traits[28].

Thirty-four vQTL associations were captured by GEIs involving age or sex. Furthermore, 391 of the 1038 GEIs (37.7%) that did not surpass conditional significance thresholds were removed after being conditioned on age or sex. Most of these associations involved phenotypes with clear sex differences. For example, circulating levels of PAEP (progestogen-associated endometrial protein, or glycodelin) showed an initial GEI with body weight. A GEI effect was not present in males or females when considered separately, but was detected when both were considered together. It was observed only because males had both higher PAEP levels and higher body weights at the minor T-allele of the *cis* variant rs697449, giving the appearance of a GEI (Supplementary Fig. 2). By contrast, we observed that age could mask rather than induce associations. PAEP was the most susceptible protein to this effect, and the effect was restricted to female participants. Negative correlations between PAEP levels and body weight were observed across genotypes in the youngest age group (40-50 years). Positive correlations were observed in the oldest age group (60−70 years, Fig. 5a). These opposing associations masked one another in the main analyses. In terms of clinical relevance, we also detected negative associations between PAEP levels and body weight in those who self-reported not having experienced menopause (mean age = 46.3, sd = 4.4 years, Fig. 5b). There were positive correlations in those who had experienced menopause (mean age = 60.5, sd = 5.4 years, Fig. 5b), and in those who had undergone a hysterectomy (mean age = 58.7, sd = 6.8 years, Supplementary Fig. 3). In all instances, the associations were strongest in carriers of the T-allele. Therefore, the allele underscored a stronger negative relationship between its protein and body composition before menopause, and a stronger positive relationship following menopause, when compared to the opposite allele. We also observed that PAEP levels were, on average, lower in older participants and by extension, those who had experienced menopause, mammograms and hormonal replacement therapy (Supplementary Figs. 4−6). Body weight was not altered by age (Supplementary Fig. 7). Together, these lines of evidence suggest that the role of glycodelin in the homoeostatic landscape may become altered in the context of menopause.

## Gene-environment interactions capture why some vQTLs lack genetic main effects on protein levels

vQTLs may lack genetic main effects if the variant positively correlates with protein levels in one stratum of an environmental exposure and negatively in other strata. The opposing effect sizes preclude a marginal main effect and produce a 'directionally discordant' GEI[17]. A slightly higher proportion of vQTLs without main effects at $p < 3.4 \times 10^{-11}$ participated in conditional GEIs compared to those with main effects on protein levels (14/67 or 20.9% versus 90/610 or 14.8%, respectively). However, 13 of the 14 vQTLs (92.9%) without additive main effects were involved in directionally discordant interactions, compared to only 2/90 (2.2%) of those with additive main effects (Supplementary Data 16). Indeed, directionally discordant interactions likely explained why 13 of the wider 67 vQTL-only sites lacked additive main effects. Associations between a given variant and protein level may not have been significant in each strata of an exposure. However, the requirement for a discordant interaction was that the effect size must have been opposite in direction in at least two strata. All 13 vQTL-only sites that are captured by such discordant GEIs are described in full in Supplementary Information. An illustrative example is highlighted below.

The *trans* indel in *FLT3* (rs147072313, fms-related tyrosine kinase 3) associated with the variance of its ligand (FLT3LG). The binding of FLT3LG to cell-surface FLT3 promotes monocyte proliferation (Fig. 6a)[29]. FLT3LG levels associated with a genotype-by-monocyte count interaction in this sample ($p = 9.3 \times 10^{-11}$). rs147072313 positively correlated with FLT3LG levels within those assigned to the highest tertile of monocyte counts ($\beta = 0.09$, se = 0.02, $p = 4.1 \times 10^{-6}$). However, there was a negative correlation in the lowest tertile ($\beta = −0.04$, se = 0.02, $p = 0.01$), precluding a main effect (Fig. 6b). The additive main effect was almost null ($p = 0.96$, shown previously in Fig. 2a). This illustrates how stratification across levels of an environmental exposure can prevent the observation of a genetic main effect. The exposure that was likely responsible for this stratification was delineated via our GEI analyses.

Figure 6C shows that the indel was associated with decreased variance of transformed FLT3LG levels ($p = 1.1 \times 10^{-15}$, also visualised previously in Fig. 2a). In relation to the GEI effect, it revealed a weak negative correlation between measured FLT3LG levels and monocyte count in major allele carriers (T-allele, $r_{major}$ = −0.03, 95% CI = [−0.04, −0.02], $p = 3.2 \times 10^{-6}$). A weak positive correlation was observed in heterozygotes ($r_{het}$ = 0.04, 95% CI = [0.02, 0.06], $p = 1.1 \times 10^{-4}$) and a stronger positive correlation was observed in indel homozygotes ($r_{minor}$ = 0.12, 95% CI = [0.06, 0.18], $p = 2.6 \times 10^{-4}$, Fig. 6c). This suggested that the effect of FLT3LG on monocyte count could be linked to rs147072313 genotype. However, the relationship between FLT3LG and FLT3 levels was consistent across genotypes, which suggested that ligand-receptor binding was preserved (Fig. 6d). Genotype also did not associate with mean differences in monocyte count ($\beta = −0.003$ per T-allele, se=0.002, $p = 0.13$, Fig. 6e). Therefore, the indel was not associated with an overall detrimental effect on monocyte count. The relationship was specific to monocytes when considering eight other blood cell types (Supplementary Fig. 8).

## Alternative explanations for variance QTLs underlying the plasma proteome

GEIs are sufficient but not necessary to generate a vQTL[15]. It is also not possible to draw a causal interpretation between tagged genetic variants, environmental exposures and variance heterogeneity. Furthermore, vQTL associations may reflect epistatic (or gene-gene) interactions, phantom vQTLs and artefacts of statistical transformations applied to phenotypes. To explore these alternative explanations, we first examined whether protein levels associated with an interaction between a vQTL and any other SNP located more than 10 Mb away on the same chromosome (see Methods, Supplementary Data 17). Of the 677 vQTL associations, 5 showed an epistatic interaction effect on their respective protein level at $p < 5.0 \times 10^{-8}$. This significance threshold was selected as we estimated that there were approximately 1 million independent epistasis tests across our analyses ($p < 0.05$ adjusted for 1 million tests).

Another important consideration in vQTL analyses is the relationship between the mean and the variance of phenotypes[16].

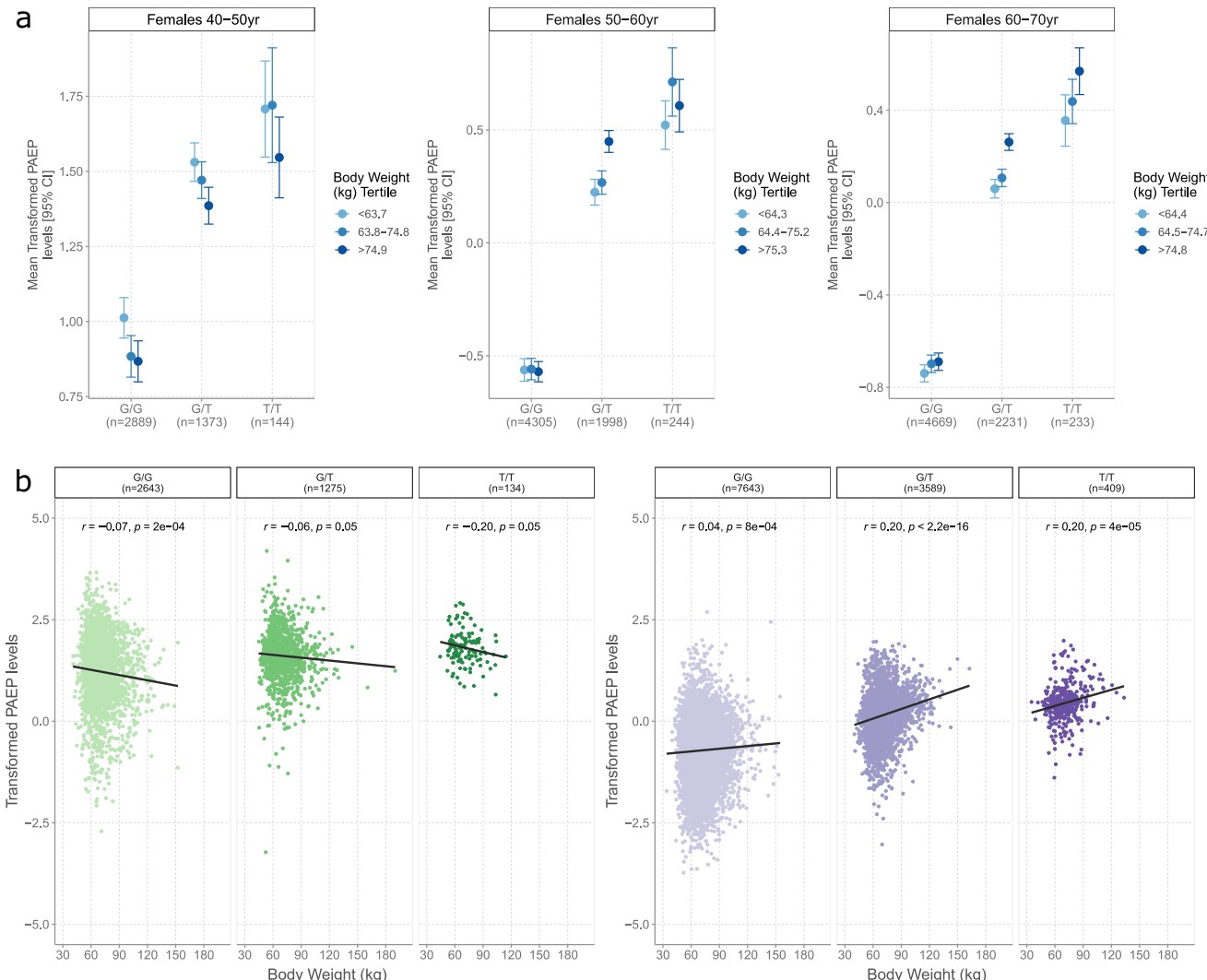

**Fig. 5 | Relationships between glycodelin levels and body weight are influenced by genotype and menopause history. a** Mean transformed glycodelin levels (PAEP, closed circles) with 95% confidence intervals (vertical bars) are shown according to tertiles of body weight (in kilograms, kg) and rs697449 genotype. Data are presented as mean values ±95% confidence intervals. The associations are stratified according to three decades of life in females. **b** Correlations between PAEP levels and body weight are shown in green (left-side) for participants who

reported 'No' to the question 'Had menopause?' at the study baseline (Field: 2724). Correlations are shown in purple for participants who reported 'Yes' to the same question. Correlations are stratified further by genotype and evidence that the T-allele underscores a stronger correlation in participants who had and also who had not experienced menopause. The statistical test used was Pearson's correlation (two-tailed).

Differences in the variance of protein levels could be explained by additional linked main effect SNPs or QTLs[24,30]. This is of particular concern for traits with QTLs of large mean-effect sizes, such as molecular phenotypes (e.g. protein levels and DNA methylation[24]), but less so for complex traits[15]. Therefore, where possible, we additionally adjusted each protein that had a vQTL association for the most significant mean-effect SNP at $p < 3.4 \times 10^{-11}$ from Sun et al.[6]. We then repeated each vQTL association test and observed that 451 (66.6% of 677) associations survived Bonferroni correction at $p < 3.4 \times 10^{-11}$ (Supplementary Data 18). Therefore, 226 associations could be captured by linkage to main effect QTLs. Of note, epistasis effects may also arise from linkage to main effect QTLs. Furthermore, these associations are indicative of phantom vQTLs but this cannot be ascertained without the use of whole genome sequencing data. Therefore, the 226 associations highlighted in these sensitivity analyses can more accurately be described as statistical artefacts. Together, 301 of all 677 vQTL associations were captured by GEIs, epistasis or statistical artefacts.

Rank-based inverse normal transformation (RINT), implemented in this study, may reduce the correlation between mean and variance effects[31] but such non-linear transformations can inflate the type I error in Levene's test in the presence of QTL effects. Therefore, as further sensitivity analyses, we re-estimated vQTL associations when preparing phenotypes in line with guidelines from Wang et al. to avoid these potential biases (see Supplementary Information)[15]. 671 variants remained at $p < 3.4 \times 10^{-11}$ with the remainder at $p < 8 \times 10^{-11}$ (Supplementary Data 19). Supplementary Information details further sensitivity analyses in which protein levels were adjusted for the exposures they associated with in stage two of our study design. vQTL association statistics were re-estimated when adjusting for participating exposures. Relative effect sizes were correlated 99% with those from the main analyses (Supplementary Data 20). Therefore, vQTL associations were largely robust to statistical transformations of phenotypes.

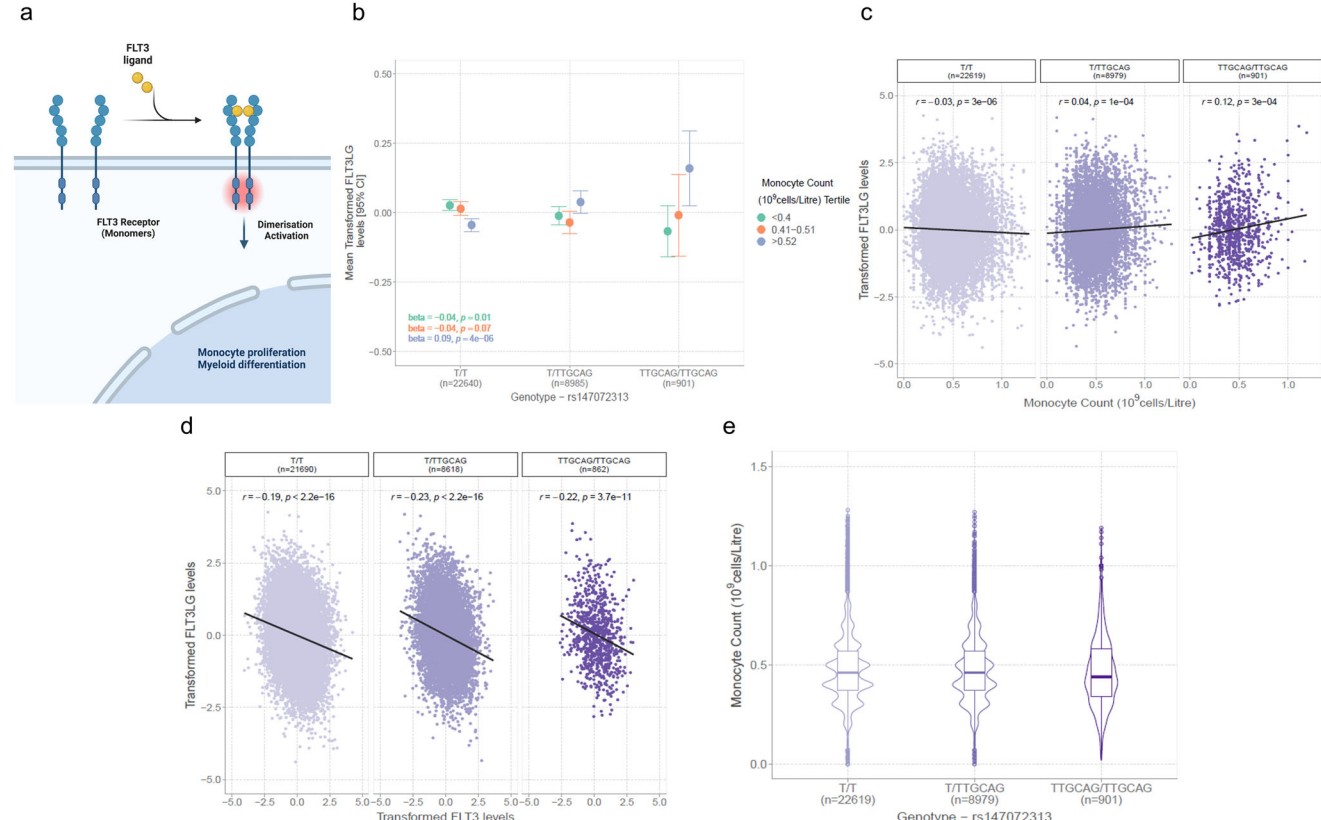

**Fig. 6 | Variance QTL in FLT3 receptor gene modifies the relationship between FLT3 ligand levels and monocyte count. a** Schematic diagram showing the role of FLT3 receptor and FLT3 ligand (FLT3LG) binding in monocyte proliferation. **b** Linear regression was used to examine the relationship between rs147072313 genotype and FLT3LG levels across tertiles of monocyte count (two-sided). Mean transformed FLT3LG levels (closed circles) with 95% confidence intervals (vertical bars) are displayed according to tertiles of monocyte count and rs147072313 genotype. Of note, the wide confidence intervals in those homozygous for the indel reflect the small sample size of this group. They represent confidence intervals for the mean of the protein levels across strata. They do not reflect the variance of the protein level, which is instead visualised in (**c**) and is most clearly illustrated via boxplots in Fig. 2a. **c** displays full distributions of FLT3LG levels and their

correlation with monocyte counts. **c** The statistical test used was Pearson's correlation (two-tailed). The correlation between transformed FLT3LG levels and monocyte count differs according to genotype at rs147072313 giving rise to a gene-environment interaction. Here, it is also apparent that the vQTL confers reduced variance in FLT3LG in carriers of the indel. **d** The association between circulating FLT3LG levels and FLT3 receptor levels is consistent across genotypes. The statistical test used was Pearson's correlation (two-tailed). (**e**) Violin plots show that rs147072313 genotype does not have an additive main effect on monocyte count. Centre line of boxplot: median, bounds of box: first and third quartiles and tips of whiskers: minimum and maximum. QTL, quantitative trait locus. Figure 6a Created with BioRender.com released under a Creative Commons Attribution-NonCommercial-NoDerivs 4.0 International license.

## Discussion

In this study, we utilised one of the world's largest proteomic datasets to perform genome-wide vQTL association studies on blood levels of approximately 1450 proteins. All vQTL associations are newly described. However, most variants showed a main effect on their protein and 30 were previously reported as lead *cis* pQTLs. We identified 1168 GEIs across 101 blood proteins. We also highlighted genotype-genotype interactions and mean-variance relationships that, together with gene-environment interactions, underpinned 301 of the 677 observed vQTL associations. Many of the GEIs identified in our study reflected known associations between protein biomarkers and environmental exposures or health outcomes (e.g. LDLR and cholesterol levels). Our data supplement the existing literature by documenting instances where the genetic control of protein levels is modified by exposures, or alternatively, where relationships between protein levels and exposures are modified by genotype. Furthermore, we catalogue a series of biologically plausible relationships that explain why some vQTL sites did not show genetic main effects on circulating protein levels. In effect, these associations reveal environmental exposures that act as stratifiers in the population and the resulting strata show apparently opposing genetic effects on protein abundances. Lastly, we detail several methodological considerations for future studies interested in

defining the GEI landscape of molecular and complex traits. Together, these data and findings bolster new toolkits in proteogenomics and impart nuanced insights that can guide biomarker discovery efforts.

Large biobank efforts have successfully detected genetic underpinnings of the human proteome, defining variants that associate with mean differences in protein levels. In terms of biological and clinical advancements, these data have supported causal inference analyses that nominate candidate drug targets for diverse disease states. A primary utility of vQTL analyses in advancing biological knowledge rests in discovering gene-environment interactions, which remain comparatively understudied. Variants with additive main effects alone (i.e. non-vQTL sites) could be used in GEI tests. However, we observed that the proportion of vQTLs participating in a GEI was five-fold higher than corresponding main effect sites, providing additional evidence that vQTLs increase the likelihood of identifying an underlying gene-environment interaction[16,17]. The higher number of main effect loci over vQTLs reflects the greater statistical power of standard regression methods when compared to Levene's test. By contrast, the power of GEI tests may increase when using vQTLs rather than main effect loci. Identifying vQTLs as a priority set of genetic variants, which can only be revealed through vQTL discovery, allows for good practice in screening for such interactions.

Our study specifically advances beyond the vQTL-GEI literature by conducting vQTL analyses on over 1400 blood proteins and considering the largest number of biomarkers in vQTL-GEI studies to date. We also detail how demographic variables (e.g. age and sex) and environmental exposures may mask or induce observed GEIs. Specific examples are discussed in detail in the following paragraphs. The findings are not readily interpreted in a clinical context given that causative pathways between the genome, proteome and phenome cannot be inferred from our computational approach alone and will require validation in mechanistic in vitro and in vivo settings and/or in targeted epidemiological studies. It is also possible that some vQTLs without GEI associations in our study are explained by exposures that we did not include. Importantly, others will be able to utilise our prioritised set of vQTLs and extract their genotypes to perform GEI tests with phenotypes of interest in their study. On balance, vQTLs should be viewed as a complementary but not competing resource to main effect QTLs in proteogenomic efforts. Our data provide new opportunities to guide studies on the safety and efficacy of candidate protein biomarkers by characterising exposures and demographic variables that impact the genetic control of circulating proteins.

GEIs identified environmental exposures and biological mechanisms that explained why some loci affect the variance of protein biomarkers only. For instance, the effect of *a trans* indel within *FLT3* on the distribution of its ligand FLT3LG was explained by an interaction with monocyte count. A conventional model suggests that lower circulating FLT3LG levels serve as a proxy for higher receptor-bound ligand. In turn, higher receptor-coupled ligand produces higher monocyte count, leading to a negative correlation with the free ligand[32]. An unexpected positive correlation was observed in indel carriers. However, monocyte count and receptor-ligand binding were stable across genotype groups. This suggested that the indel may instead impact downstream ligand-mediated signalling processes without exerting overall detrimental effects on blood cell profiles.

Age and sex showed complex effects on GEIs. A key illustrative example of their effects involved associations between glycodelin and physical traits. GEIs that were confounded by sex were attributed to sexual dimorphism in body composition and QTL effects. By contrast, associations between glycodelin and body composition were masked by age within females only. Glycodelin has four major glycoforms (-A,-S,-F and −C) with prominent roles in reproduction, pregnancy and immune function[33]. They also exhibit differential expression across sexes and tissues. Glycodelin-A is expressed in the female genital tract[34,35]. Glycodelin-F and −C are detected in follicular fluid and in cumulus cells within the ovarian follicle, respectively[36]. Glycodelin-S is secreted from seminal vesicles into seminal fluid[37,38]. Temporal studies on the expression of glycodelin are primarily restricted to pregnancy and ovulation and less is known about its characteristics throughout the lifespan[39]. We observed that glycodelin levels were lower in participants who had experienced menopause, who were on hormonal replacement therapy or who had undergone a hysterectomy. This is likely attributed to reduced levels of progesterone, which maintains glycodelin levels[40]. Body weight was not altered in our sample by age or by history of menopause. However, the correlation between glycodelin and body composition switched in direction following menopause or hysterectomy, which may suggest that its role in regulatory mechanisms becomes altered after these events. It is challenging to separate whether this is due to relevant medications or in response to biological cascades. Our data provide methodological considerations for other GEI studies and further underscore the need for large samples that can enable stratified analyses and biomarker research in understudied groups.

This study has a number of limitations. First, we note an absence of an external replication cohort. The split of the sample into discovery and replication sets was carried out in order to directly match the sample split and approach of Sun et al., who report a main effect QTL study on the same proteins. The statistical power of Levene's test is weaker than that of linear regression GWAS[14]. Therefore, it is advantageous to implement a large available sample size in vQTL association studies. Presently there is no external cohort with the same proteins measured, with matched phenotype definitions for GEI analyses and is of sufficient sample size to permit an informative external replication analysis. Hence, we elected to retain the analytical approach of Sun et al. Furthermore, while we observed evidence for ancestry-specific effects in some vQTL association tests, larger sample sizes will also be needed to provide meaningful, fine-scale ancestry-specific and cross-ancestry vQTL discovery efforts. Second, on a related note, the majority of participants were of European ancestry. Therefore, it is not possible to clearly generalise gene-environment effects to other ancestry groups given potential differences in both genetic and environmental profiles. Third, specific quality control parameters could not feasibly be applied to all phenotypes tested. Additional confounders and phenotype-specific transformations may need to be considered in follow-up or mechanistic studies. Complex interactions between multiple environmental exposures or inaccurately recorded phenotypes could also have precluded GEI detection within the large correlated constellation of phenotypes studied.

The study complements existing proteogenomic efforts by considering additional distributional properties of the proteome whilst cataloguing biologically informative examples of its interaction with the genome, metabolome and phenome. Our datasets of variance QTL effects and GEIs establish unmet resources in the pursuit of accelerating biomarker discovery and validation.

## Methods

### UK biobank study

UK Biobank, or UKB, is a prospective, population-based cohort of approximately 500,000 individuals aged between 40 and 69 years at recruitment[41]. Recruitment took place between 2006 and 2010. Here, a subset of the UKB sample was utilised, which was defined by The UKB Pharma Proteomics Project or UKB-PPP consortium. The consortium comprises 13 biopharmaceutical companies, which funded the generation of blood-based proteomic data. The UKB-PPP sample includes 54,219 participants and consists of (i) a randomised subset of 46,595 UKB participants at the baseline visit, (ii) 6376 individuals at the baseline selected by the UKB-PPP consortium members and (iii) 1268 individuals who participated in the COVID-19 repeat imaging study.

### Protein measurement in UK Biobank

Blood samples from 54,219 UKB-PPP participants were analysed using the Olink Explore 1536 platform. The platform uses Proximity Extension Assay[42] and measured 1,472 protein analytes across four Olink panels (Cardiometabolic, Inflammation, Neurology and Oncology). The analytes reflect 1,463 unique proteins. EDTA-treated plasma samples (60 μl) were serially diluted to 1:10, 1:100 and 1:1000 and transferred to 384-well plates. Samples were processed in eight batches (termed batches 0-7) and incubated with antibodies overnight at −4 °C. Olink's inbuilt quality control (QC) workflow returned Normalized Protein eXpression (NPX) values, which is a relative quantification unit on a log-2 scale. Full details on protein measurement and QC are available in Supplementary Information.

### Genotyping in UK Biobank

The UKB genotype dataset includes 488,377 participants. Of these, 49,950 individuals were genotyped on the Applied Biosystems UK BiLEVE Axiom™ Array and 438,427 participants were genotyped on the closely related UK Biobank Axiom™ Array[41]. Here, we followed the genotype QC process of Sun et al.[6] in order to enable direct comparisons to a conventional main effect QTL analysis using the same sample[6]. Briefly, UKB genotype data were imputed to the Haplotype Reference Consortium[43] and UK10K[44] reference panels. Imputed

genetic variants were filtered for INFO > 0.7 and minor allele count>50, and chromosome positions were lifted to hg38 build using LiftOver[45]. Variants with a genotyping rate >99%, Hardy-Weinberg equilibrium test $p > 10^{-15}$ and <10% missingness were retained. Sun et al.[6] utilised variants with minor allele frequency (MAF) > 1%. We applied a higher threshold of MAF > 5% in accordance with the workflow of Wang et al. for vQTL analyses[15]. Following QC, 6,815,338 variants remained. There were 52,363 individuals with paired genotype and protein data following QC protocols.

The UKB-PPP sample was separated into discovery ($n = 34,557$) and replication subsets ($n = 17,806$) as per the design of Sun et al.[6]. The discovery set included participants who were of European ancestry and present in Olink measurement batches 1–6. The remaining samples comprised the replication set and included 10,840 White, 931 African, 920 Central/South Asian, 308 Middle Eastern, 262 East Asian, and 97 admixed American ancestries.

## Variance QTL association studies
vQTL association studies were performed using the vQTL suite in OSCA (version 0.46)[46]. Levene's test with median was applied. The false-positive rate of this test has been shown to be well-calibrated across simulated data in comparison to other commonly-used vQTL methods[15].

In the discovery cohort, rank-based inverse normal transformed protein values (NPX) were regressed onto age, $age^2$, sex, age*sex, $age^2$*sex, batch, UKB study centre, UKB genotype array, time between blood sampling and measurement and 20 genetic principal components. One additional covariate was included in the replication set, which indicated whether samples were pre-selected by consortium members or as part of the COVID imaging study. Summary data for covariates are available in Supplementary Data 21 and their associations with protein levels are shown in Supplementary Data 22 and 23 for discovery and replication sets, respectively. The preparation of protein data was aligned as closely as possible to the corresponding main effect QTL study by Sun et al.[6]. Of note, Sun et al. did not adjust protein levels prior to genetic association studies and instead included fixed-effect covariates. This was not possible in our study as the Levene's test module did not permit fixed-effect covariates. Residuals were standardised to Z-scores and entered as dependent variables. Additively-coded genotype status was included as the independent variable.

A Bonferroni-corrected significance threshold of $p < 3.4 \times 10^{-11}$ was applied ($p < 5 \times 10^{-8}$ adjusted for 1463 proteins). Primary associations were defined by clumping variants ±1 Mb around significant vQTLs using PLINK (version 1.9)[47] with the exception of the HLA locus (chromosome 6: 25.5–34.0 Mb). The larger HLA locus was considered as one region due to its complex linkage disequilibrium structure.

## Annotation of vQTLs
We deemed vQTLs to be *cis* sites if they were within 1 Mb from the canonical transcription start site of the gene encoding the respective protein. A vQTL was allocated as a *trans* site if it fell outwith this region. *Trans* vQTLs can point towards hierarchical networks of genes within regulatory pathways. They may also reflect epistatic interactions with *cis* variants or capture networks of proteins that are similarly impacted by a given GEI. Genomic annotation was performed using two strategies. In the first strategy, annotation was carried out using Ensembl Variant Effect Predictor (VEP) (version 110), ANNOVAR and (https://annovar.openbioinformatics.org/en/latest/) and WGS Annotator (WGSA, https://sites.google.com/site/jpopgen/wgsa) (version 0.95). Gene annotation was based on RefSeq and Ensembl. The rank of genic intolerance was estimated for synonymous mutations along with the consequent susceptibility to disease based on the ratio of loss-of-function. SIFT and PolyPhen scores for changes to protein sequence were estimated for coding variants. For non-coding variants, transcription factor binding site, promoters, enhancers and open chromatin regions were mapped to histone marks chip-seq, ATAC-seq and

DNase-seq data from The Encyclopaedia of DNA Elements Project (ENCODE, https://www.encodeproject.org) and ROADMAP Epigenomics Mapping Consortium (http://www.roadmapepigenomics.org). For intergenic variants, the 5' and 3' nearby protein coding genes were mapped and the distance between the 5' transcription starting sites of a protein coding gene to the variant was provided. In the second strategy, variants were submitted to the Open Targets Genetics platform to systematically estimate to most likely gene annotation for a given vQTL[21,22]. A variant-to-gene score is estimated as an aggregate score based on several lines of evidence from functional genomics data (e.g. chromatin conformation, chromatin interactions) and quantitative trait loci (expression, protein and splicing QTLs). This was carried out as the gene most functionally implicated by a given variant is not strictly that which holds the shortest distance between the variant and its canonical TSS. Gene annotation is presented for both strategies in Supplementary Data 2, which agreed for 55% of variants.

## Preparation of phenotypes for gene-environment interaction tests
First, we aimed to identify phenotypes of interest in a systematic manner. vQTL variants were queried against the GWAS Catalogue[25] and traits which associated with the variants at a conventional significance threshold of $p < 5 \times 10^{-8}$ were extracted. The list of associated traits was then queried against the UK Biobank data resource to assess if they were measured or assayed across our participants. This allowed us to include exposures that had known main effect associations with our set of genetic variants. Second, we considered remaining quantitative lifestyle and metabolic factors available in the UK Biobank for GEI analyses, given prior evidence for interactions between the proteome and metabolome[48,49]. Third, we identified additional continuous phenotypes from a recent protein vQTL analysis by Westerman et al.[17] and ensured no technical QC variables were considered. Prior to QC, 782 candidate phenotypes were deemed to be relevant and were considered for GEI tests. Phenotypes with excessive missingness (i.e. <5% complete data, $n = 229$) and insufficient heterogeneity ($n = 35$) were removed. There were 518 phenotypes following QC. Categorical variables were converted into binary or ordinal phenotypes as appropriate. Continuous variables were rank-based inverse normal transformed. The phenotypes or 'exposures' consisted of 113 psychological, 103 physical, 103 biochemical, 83 health and disease, 79 dietary, 36 lifestyle and one technical variable (season of blood draw). Further information on phenotype preparations is available in Supplementary Information.

## Gene-environment interaction tests
Linear regression models tested whether the levels of a given protein were associated with an interaction between its vQTL (stage one) and each of 518 possible exposures (stage two):

$$\text{Protein levels(standardised residuals)} \sim \text{SNP(0,1,2)*exposure + SNP + exposure}$$

(1)

No fixed-effect covariates such as age and sex were included in GEI association models as protein levels were already corrected for relevant covariates in stage one, and in keeping with prior GEI studies[17]. Of note, the protein variable was entered as the dependent variable to align with the previous literature[17]. Further, this was carried out as we were interested in uncovering genotype-dependent correlations between protein levels and environmental factors. We do not attempt to draw causal conclusions from these interaction effects regarding pathways connecting genetic variation, protein levels and environmental variables. A Bonferroni-corrected threshold of $p < 1.4 \times 10^{-7}$ was applied ($p < 0.05$ adjusted for 677 vQTLs and 518 exposures). Conditional GEI tests were performed in stage two in order to further account for the correlation structure between related phenotypes. All GEIs for a given protein that withstood multiple testing correction were brought

forward to stepwise conditional analyses. GEIs for a given protein were re-tested while iteratively conditioning on the association that involved the most significant exposure (i.e. smallest *p* value for a given protein). The process was repeated until no further GEIs could be considered. As described in Weller et al., we ensured to account for covariate × environment and covariate × gene interaction terms in order to account for the possible effect of confounding variables[50].

## Inclusion & Ethics

All participants provided informed consent. This research has been conducted using the UK Biobank Resource under approved application numbers 65851, 20361, 26041, 44257, 53639, 69804.

## Reporting summary

Further information on research design is available in the Nature Portfolio Reporting Summary linked to this article.

## Data availability

The genome-wide vQTL and GEI summary statistics generated in this study have been deposited in the Synapse database under the following https://doi.org/10.7303/syn61514369. The underlying NPX measures are available through the UK Biobank Research Analysis Portal (https://www.ukbiobank.ac.uk/enable-your-research).

## Code availability

All code is available with open access at the following GitHub repository: https://github.com/robertfhillary/vqtls-uk-biobank [51].

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

## Acknowledgements

We thank the participants, contributors, and researchers of UK Biobank for making data available for this study – with special thanks to Lauren Carson, John Busby, Naomi Allen and Rory Collins for making the study possible. We are grateful to the research & development leadership teams at the thirteen participating UKB-PPP member companies (Alnylam Pharmaceuticals, Amgen, AstraZeneca, Biogen, Bristol-Myers Squibb, Calico, Genentech, Glaxo Smith Klein, Janssen Pharmaceuticals, Novo Nordisk, Pfizer, Regeneron, and Takeda) for funding the study. We thank the Legal and Business Development teams at each company for overseeing the contracting of this complex, pre-competitive collaboration—with particular thanks to Erica Olson of Amgen, Andrew Walsh of GSK and Fiona Middleton of AstraZeneca. Finally, we thank the team at Olink Proteomics (Philippa Pettingell, Klev Diamanti, Cindy Lawley, Linda Jung, Sara Ghalib, Ida Grundberg and Jon Heimer) for their consistent logistic support throughout the project—with special thanks to Evan Mills for co-championing the project and leading internal activities at Olink.

An Alzheimer's Society major project grant (AS-PG-19b-010 to R.E.M.) supported this work. D.A.G. was supported by the Wellcome Trust Translational Neuroscience programme (108890/Z/15/Z to D.A.G.). R.F.H. was supported by a British Heart Foundation Immediate Fellowship (FS/IPBSRF/22/27042 to R.F.H.).

## Author contributions

R.F.H., D.A.G., Z.K., T.M., H.R., Biogen Biobank team, R.E.M., C.N.F. and B.B.S. conceptualised the study design and consulted on methods and results. R.F.H. performed all analyses. T.L., K.F. and H.M. performed quality control or prepared the proteomics dataset. All authors reviewed and approved of the manuscript.

## Competing interests

T.L., K.F., H.M., H.R. and B.B.S. are employed by Biogen. Z.K. and C.N.F. are employed by Optima Partners. R.F.H., D.A.G. and R.E.M. act as scientific consultants for Optima Partners. R.E.M. is an advisor to the Epigenetic Clock Development Foundation and has received speaker fees from Illumina. R.F.H. has received consultant fees from Illumina. All other authors declare no competing interests.

## Additional information

## Biogen Biobank Team

Eric Marshall[3], Tinchi Lin[3], Kyle Ferber[3], Helen McLaughlin[3], Heiko Runz[3] & Benjamin B. Sun [3,5,6] ✉

