## [Peer Review File · Nature Communications]

Systematic discovery of gene-environment interactions underlying the human plasma proteome in UK BiobankEditorial Note: This manuscript has been previously reviewed at another journal that is not operating a transparent peer review scheme. This document only contains reviewer comments and rebuttal letters for versions considered at *Nature Communications*.

REVIEWER COMMENTS

Reviewer #1 (Remarks to the Author):

I would like to thank the authors for their excellent job in addressing my comments. I have no further comments. I'm happy with the paper to be published.

Reviewer #4 (Remarks to the Author):

Hillary et al. have (again) done a great effort in addressing my concerns, and I would recommend acceptance pending two minor remaining concerns. Both relate to adjustments to test for 'independent' interaction effects. That is in detail described here [doi:10.1016/j.biopsych.2013.09.006](https://doi.org/10.1016/j.biopsych.2013.09.006).

1) I do not agree with the strategy presented to rule out that proteins with multimodal distributions because of strong cis-pQTLs do not introduce artificial variance trans-pQTLs. Simply adjusting (or regressing out) the effect of the cis-pQTL will not be sufficient. This assumes an effect of the cis-variant that is consistent for each allele interacting with 'E'. The authors would need to add an interaction term (+plus main effects) between the cis-pQTL and the 'E' when testing the trans-pQTL x 'E' interaction.

2) The same applies to the section that tries to tease apart the most important 'E' for each of the vpQTLs.

We are very grateful for the comments provided by the editor and each of the external reviewers of this manuscript. Please see below, in blue, a detailed response to the comments and remaining concerns raised by Reviewer #4. We hope that remaining concerns are adequately addressed in the revised manuscript.

Reviewer #4:

Comment #1

“Hillary et al. have (again) done a great effort in addressing my concerns, and I would recommend acceptance pending two minor remaining concerns. Both relate to adjustments to test for ‘independent’ interaction effects. That is in detail described here doi:10.1016/j.biopsych.2013.09.006.

I do not agree with the strategy presented to rule out that proteins with multimodal distributions because of strong cis-pQTLs do not introduce artificial variance trans-pQTLs. Simply adjusting (or regressing out) the effect of the cis-pQTL will not be sufficient. This assumes an effect of the cis-variant that is consistent for each allele interacting with ‘E’. The authors would need to add an interaction term (+plus main effects) between the cis-pQTL and the ‘E’ when testing the trans-QTL x ‘E’ interaction.”

Response: We thank the reviewer for their kind and constructive comments on our manuscript. We also thank the reviewer for highlighting the above publication and their helpful feedback on sections of the paper involving interaction effects. In light of this, we agree that the previous approach would have been insufficient in that it would have assumed a consistent effect of a main-effect *cis* QTL across all alleles that interact with ‘E’.

We first wish to highlight that in our previous response, we only tested for the influence of *cis* pQTLs at the discovery vQTL stage. This occurred before any tests with environmental variables. Hence, we had not incorporated information on environmental variables. As a brief reminder, 100 of 204 trans vQTLs were available for that sensitivity analysis (also having a lead *cis* pQTL with complete genotype data). 88 remained associated with their respective protein at a Bonferroni-corrected significance threshold of $p < 3.4 \times 10^{-11}$ with the remainder present at $p < 5.0 \times 10^{-6}$. We now extend this to involve the *trans*-QTL * ‘E’ interaction. Of the 100 *trans* vQTLs with corresponding *cis* QTLs, we extracted those with a partner ‘E’ variable in GEI tests. We repeated such GEI tests whilst accounting for an interaction term (and main effects) between the *cis* pQTL and ‘E’ variables as indicated by the reviewer. We identified 36 associations that satisfied this criteria i.e. involved a *trans* vQTL and ‘E’, and had a *cis* pQTL available for testing. All survived the original Bonferroni correction threshold when additionally modelling an interaction between the *cis* pQTL and ‘E’. All of the *trans* vQTLs would have survived the discovery stage when accounting for these variables.

Comment #2

“The same applies to the section that tries to tease apart the most important ‘E’ for each of the vpQTLs.”

Response: We are pleased to report that our approach matches that described in Keller *et al.* (doi:10.1016/j.biopsych.2013.09.006), which provides confidence in our methodology. Indeed, to account for potential confounders, we modelled covariate × environment and covariate × gene interaction terms in attempts to tease out the most important ‘E’ for each vpQTL. However, we appreciate that this was not made sufficiently clear in our initial descriptions. We provide revised text as follows on lines 722-726:

“GEIs for a given protein were re-tested while iteratively conditioning on the association that involved the most significant exposure (i.e. smallest p -value for a given protein). The process was repeated until no further GEIs could be considered. As described in Weller *et al.*, we ensured to account for covariate × environment and covariate × gene interaction terms in order to account for the possible effect of confounding variables⁵⁶.”

REVIEWERS' COMMENTS

Reviewer #4 (Remarks to the Author):

I thank the authors for addressing my remaining methodological concerns, and congratulate to a great body of work.